# Heterochromatin-dependent transcription of satellite DNAs in the *Drosophila melanogaster* female germline

Xiaolu Wei[1]*, Danna G Eickbush[2], Iain Speece[2], Amanda M Larracuente[2]*

[1]Department of Biomedical Genetics, University of Rochester Medical Center, Rochester, United States; [2]Department of Biology, University of Rochester, Rochester, United States

**Abstract** Large blocks of tandemly repeated DNAs—satellite DNAs (satDNAs)—play important roles in heterochromatin formation and chromosome segregation. We know little about how satDNAs are regulated; however, their misregulation is associated with genomic instability and human diseases. We use the *Drosophila melanogaster* germline as a model to study the regulation of satDNA transcription and chromatin. Here we show that complex satDNAs (>100-bp repeat units) are transcribed into long noncoding RNAs and processed into piRNAs (PIWI interacting RNAs). This satDNA piRNA production depends on the Rhino-Deadlock-Cutoff complex and the transcription factor Moonshiner—a previously described non-canonical pathway that licenses heterochromatin-dependent transcription of dual-strand piRNA clusters. We show that this pathway is important for establishing heterochromatin at satDNAs. Therefore, satDNAs are regulated by piRNAs originating from their own genomic loci. This novel mechanism of satDNA regulation provides insight into the role of piRNA pathways in heterochromatin formation and genome stability.

*For correspondence:
xiaolu_wei@urmc.rochester.edu (XW);
alarracu@bio.rochester.edu (AML)

**Competing interests:** The authors declare that no competing interests exist.

## Introduction

Repetitive DNA makes up a large fraction of eukaryotic genomes (*Britten and Kohne, 1968*). Most repeat-dense genomic regions are gene-poor and tightly packed into heterochromatin (reviewed in *Allshire and Madhani, 2018*; *Janssen et al., 2018*). Tandem arrays of repeated sequences called satellite DNAs (satDNAs) are abundant in the heterochromatin of the pericentromeres, subtelomeres, and on sex chromosomes (*Charlesworth et al., 1994*; *Schmidt, 1998*; *Schueler et al., 2001*). SatDNAs are typically viewed as selfish genetic elements that can spread rapidly in genomes and are generally repressed (*Doolittle and Sapienza, 1980*; *Orgel and Crick, 1980*). The de-repression of satDNA is associated with cellular senescence and various cancers (e.g., *Ting et al., 2011*; *Zhu et al., 2011*). However, satDNAs play roles in chromatin structure, chromosome segregation, and genome stability across a wide range of taxa (*Weiler and Wakimoto, 1995*; *Dernburg et al., 1996*; *Lippman et al., 2004*; *Bouzinba-Segard et al., 2006*; *Zhu et al., 2011*; *Swanson et al., 2013*; *Plohl et al., 2014*; *Rošić et al., 2014*). SatDNA-derived transcripts have been detected in many species (*Ugarkovic, 2005*; *Usakin et al., 2007*; *Biscotti et al., 2015*; *Ferreira et al., 2015*). In insects, these transcripts may have roles in early embryos (*Pathak et al., 2013*; *Halbach et al., 2020*) and spermatogenesis (*Mills et al., 2019*). Across organisms, satDNA-derived transcripts may generally be important for maintaining genome stability and integrity, yet the regulation and function of these transcripts remains poorly understood (reviewed in *Janssen et al., 2018*).

Insights might come from the small RNA pathways that protect genome integrity by silencing repeats. These RNA interference pathways play roles in heterochromatin formation and maintenance at repeats across species (*Hall et al., 2002*; *Volpe et al., 2002*; *Fukagawa et al., 2004*;

*Noma et al., 2004*; *Verdel et al., 2004*; *Novo et al., 2020*). In these pathways, small RNAs guide Argonaute proteins to cleave mRNA or silence genomic DNA by targeting complementary sequences (*Hutvagner and Simard, 2008*). Among the most abundant types of repeat-derived small RNAs in animal germlines are the 23–32-nt PIWI-interacting RNAs (piRNAs) that target transposable elements (TEs)—genomic parasites that mobilize and can cause genome instability (*Aravin et al., 2006*; *Girard et al., 2006*; *Grivna et al., 2006*; *Lau et al., 2006*; *Brennecke et al., 2007*; *Houwing et al., 2007*). These piRNAs are particularly well-studied in *Drosophila* ovaries. The piRNA precursors are transcribed from discrete genomic loci containing primarily truncated TE sequences, called piRNA clusters. The piRNAs derived from these loci repress TE activity through both post-transcriptional (*Brennecke et al., 2007*; *Gunawardane et al., 2007*) and transcriptional silencing. In ovaries, piR-NAs guide Piwi to genomic locations with complementary nascent RNAs and recruit heterochromatin factors to silence TEs (*Wang and Elgin, 2011*; *Sienski et al., 2012*; *Le Thomas et al., 2013*; *Rozhkov et al., 2013*).

There are two main types of piRNA sources in *Drosophila* ovaries—uni-strand and dual-strand piRNA clusters. Uni-strand piRNA clusters require promoter sequences and are either expressed only in somatic tissues (e.g., *flamenco*) or in both somatic tissues and the germline (e.g., *20A*) (*Brennecke et al., 2007*; *Malone et al., 2009*; *Mohn et al., 2014*). However, most piRNA clusters are heterochromatic dual-strand clusters, which are bidirectionally transcribed and do not necessarily require promoters (e.g., *42AB*, *80F*, and *38C1/2*; *Brennecke et al., 2007*; *Mohn et al., 2014*; *Andersen et al., 2017*). Dual-strand piRNA clusters are expressed primarily in the germline, where their transcription is licensed by a non-canonical pathway that depends on the heterochromatin protein-1 (HP1) variant called Rhino (Rhi) (*Klattenhoff et al., 2009*; *Zhang et al., 2014*). Rhi recruits Deadlock (Del), an unstructured linker protein (*Wehr et al., 2006*; *Czech et al., 2013*), and Cutoff (Cuff), a protein related to the yeast Rai1 decapping enzyme (*Pane et al., 2011*), to H3K9me3 chromatin. This complex is referred to as the Rhino, Deadlock, and Cutoff (RDC) protein complex (*Mohn et al., 2014*). Moonshiner (Moon)—a paralog of the transcription factor TFIIA-L—interacts with Del and recruits TBP-related factor 2 (TRF2) to initiate transcription of dual-strand piRNA clusters (*Andersen et al., 2017*). Most piRNA studies in *Drosophila* focus on their important role in repressing TE activity to protect genome integrity (e.g., *Brennecke et al., 2007*). Given that TEs and satDNAs both are abundant repeats in heterochromatin whose activities are associated with genomic instability, we suspect that satDNAs may also be regulated by this piRNA pathway.

Consistent with our hypothesis, small RNAs derived from satDNAs exist in germlines (e.g., *Aravin et al., 2003*; *Saito et al., 2006*). However, little is known about these satDNA-derived small RNAs. Here we leverage publicly available RNA-seq and ChIP-seq datasets and complement these data with cytological and molecular analyses of expression to study the regulation of satDNAs in the germline. SatDNAs are categorized based on their repeat unit size as simple (1–10 bp) or complex (>100 bp). We focus on two abundant families of complex satDNA in *Drosophila melanogaster*: *Responder* (*Rsp*) and satellites in the *1.688* g/cm$^3$ family (*1.688*). We show that complex satDNAs are expressed and processed primarily into piRNAs in both testes and ovaries. In ovaries, this expression depends on the RDC complex and Moon. Disruptions of the piRNA pathway lead to a loss of both satDNA-derived piRNAs and heterochromatin marks at satDNA loci. Our analyses suggest a model where the establishment of heterochromatin at satDNA is regulated by piRNAs originating from their own genomic loci. These findings add insight into the role of piRNA pathways in heterochromatin formation and genome stability.

## Results and discussion

### SatDNA transcripts originate primarily from large heterochromatic satDNA blocks

To study satDNA expression patterns, we characterized transcripts from two representative complex satDNA families in *D. melanogaster*—*Rsp* and *1.688*—across tissues and developmental time points. *Rsp* consists of a dimer of two closely related ~120 bp repeats in the pericentric heterochromatin on chromosome *2R* of *D. melanogaster* (*Wu et al., 1988*; *Pimpinelli and Dimitri, 1989*). The *1.688* family of repeats is the most abundant complex satDNA in *D. melanogaster* (*Lohe and Roberts, 1988*). It comprises different subfamilies that exist as discrete tandem arrays in the pericentric

heterochromatin named after their repeat unit sizes on chromosome *2L* (*260-bp*), chromosome *3L* (*353-bp* and *356-bp*), and the X chromosome (*359-bp*) (*Losada and Villasante, 1996*; *Abad et al., 2000*). Because there is high sequence similarity among these repeats, we analyzed all *1.688* subfamilies together unless stated otherwise.

We mined modENCODE datasets (*Supplementary file 1* and *Graveley et al., 2011*; *Brown et al., 2014*) and found evidence for satDNA expression in total RNA-seq datasets from both sexes and across different developmental stages (*Figure 1*, *Figure 1—figure supplement 1*). Both satDNA families are expressed in gonads, head, and other tissues (*Figure 1A*, *Figure 1—figure supplement 1C*). Their transcript abundance is low ($RPM_{Rsp} < 10$ and $RPM_{1.688} < 300$; *Supplementary file 2*) and generally increases throughout development and with adult age (*Figure 1—figure supplement 1A, B*). SatDNA-derived reads have very low abundance in the poly-A selected RNA-seq data ($RPM_{Rsp} < 0.2$ and $RPM_{1.688} < 10$; *Supplementary file 2*), indicating that the majority of satDNA transcripts are not polyadenylated.

To validate the presence of satDNA-derived transcripts in gonads, we used RNA fluorescence in situ hybridization (FISH). Both *Rsp* and *1.688* satellite transcripts are visible in testes and ovaries (*Figure 1B*, *Figure 1—figure supplement 2A*). These signals are undetectable after treating with

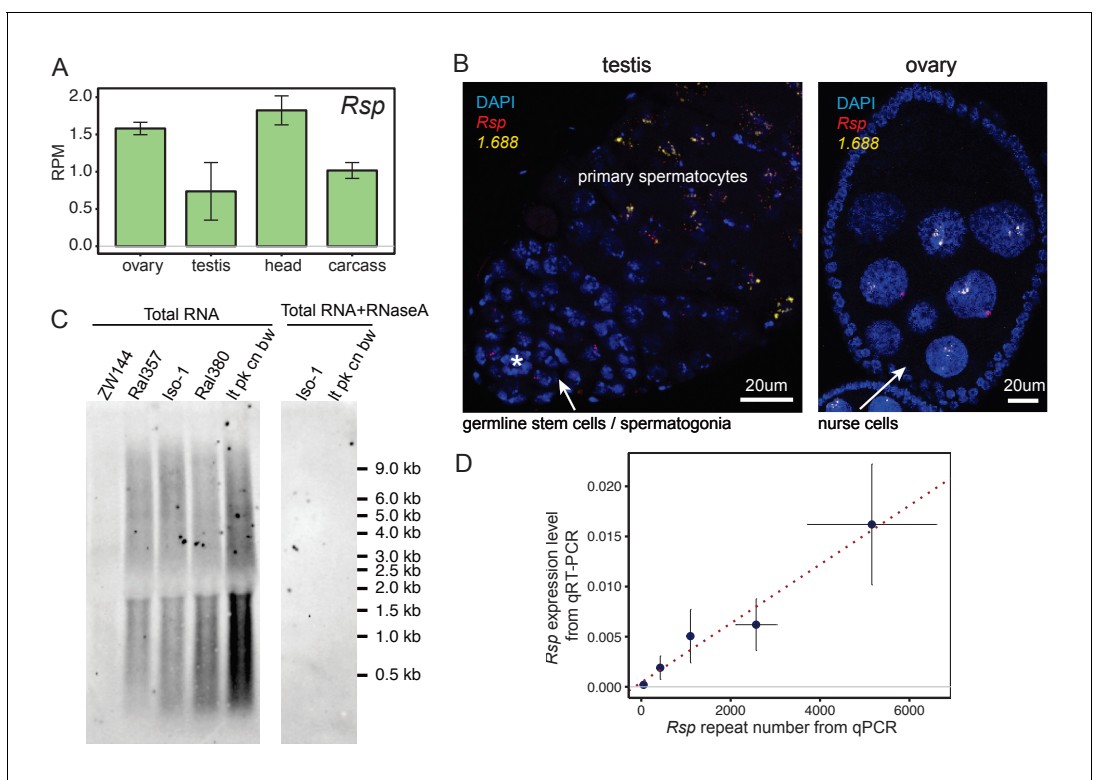

**Figure 1.** Satellite DNAs (satDNAs) are expressed in ovaries and testes. (A) *Rsp* satDNA transcription level in various tissues (corresponding result for *1.688* is shown in *Figure 1—figure supplement 1C*). Carcass: whole body without the head, reproductive organs, and digestive tract. Source data in *Figure 1—source data 1*. (B) RNA fluorescence in situ hybridization shows evidence for *Rsp* and *1.688*-derived transcripts in testes and ovaries; asterisk indicates the hub. The probe for *1.688* recognizes all *1.688* subfamilies except for *260-bp* on chromosome *2L*. (C) Northern blot probed with *Rsp*. Total RNA was extracted from ovaries of fly lines with varying copy numbers of *Rsp*: ZW144 (200 copies), Ral357 (600 copies), Iso-1 (1100 copies), Ral380 (2300), and *lt pk cn bw* (4100). There is no signal after RNaseA treatment. Signal quantification (shown in *Figure 1—figure supplement 1D*) shows *Rsp* transcript abundance correlates with its genomic copy number (Pearson's correlation coefficient $r^2 = 0.93$, p-value=0.02). (D) qPCR and qRT-PCR quantification of *Rsp* copy number and expression level, respectively, of strains used in northern blot. A linear regression line is shown in the plot with red dotted line (Pearson's correlation coefficient $r^2 = 0.98$, p-value=0.003). Details for (C) and (D) in *Supplementary file 4*.

The online version of this article includes the following source data and figure supplement(s) for figure 1:

**Source data 1.** Satellite DNAs (SatDNA) transcription level in various developmental stages and tissues.

**Figure supplement 1.** Satellite DNAs (satDNAs) are expressed across tissues and developmental stages.

**Figure supplement 2.** RNA fluorescence in situ hybridization (FISH) signals of satellite DNAs (satDNAs) are from RNAs, not DNAs.

RNaseA prior to probe hybridization (*Figure 1—figure supplement 2B*), which degrades single-stranded RNAs, or RNaseH post-probe hybridization (*Figure 1—figure supplement 2C*), which degrades the RNA in DNA-RNA hybrids. This suggests that these signals are from RNA rather than DNA. We detected satDNA transcript foci in ovarian nurse cells and in pre-meiotic testicular germ cells. Interestingly, in testes we detected *Rsp* signal at earlier stages of spermatogenesis (i.e., germ-line stem cells/spermatogonia) than the *1.688* signals (i.e., primary spermatocytes; *Figure 1B*). The difference in timing is notable as *Rsp* is the specific target of *Segregation Distorter* (*SD*; *Sandler et al., 1959*): a well-known male meiotic drive system that causes a defect in post-meiotic germ cells (reviewed in *Larracuente and Presgraves, 2012*). *Rsp* transcription may therefore play some specific role in the male germline distinct from other complex satDNA.

The bulk of satDNAs are found in large blocks of tandem repeats in the heterochromatin with small blocks occurring in the euchromatin (*Waring and Pollack, 1987*; *DiBartolomeis et al., 1992*; *Kuhn et al., 2012*; *Sproul et al., 2020*). Some of the euchromatic (*Menon et al., 2014*; *Joshi and Meller, 2017*; *Deshpande and Meller, 2018*) and heterochromatic loci in the *1.688* family (*Usakin et al., 2007*) are transcribed. To determine if satDNA-derived transcripts originating from large heterochromatic loci is a general feature of other complex satDNAs, we examined transcript size and abundance in total RNA from ovaries of flies that vary in *Rsp* repeat copy number (*Supplementary file 3*; *Khost et al., 2017*). We determined that, while transcript lengths were similar among these lines—ranging between <300 nt and >9000 nt (*Figure 1C*)—the abundance of *Rsp* transcripts correlated with genomic copy number (*Figure 1—figure supplement 1D* and *Supplementary file 4*, Pearson's correlation coefficient $r^2 = 0.93$, p-value=0.02). We validated these hybridization results using qPCR and qRT-PCR to quantify *Rsp* genomic DNA and RNA transcript abundance, respectively (*Figure 1D* and *Supplementary file 4*, Pearson's correlation coefficient $r^2 = 0.98$, p-value=0.003). The correlation between genomic copy number and transcript abundance is consistent with most transcripts originating from the large blocks of heterochromatic satDNA.

## SatDNA transcripts are processed into piRNAs in *Drosophila* germline

Many different repeat-derived transcripts are processed into piRNAs (*Aravin et al., 2003*; *Saito et al., 2006*; *Brennecke et al., 2007*) and endo-siRNAs (*Czech et al., 2008*; *Ghildiyal et al., 2008*; *Okamura et al., 2008*; *Menon et al., 2014*). To ask if complex satDNA-derived RNAs are processed into small RNAs, we reanalyzed published small RNA-seq data (*Supplementary file 1*; *Ghildiyal et al., 2010*; *Rozhkov et al., 2010*; *Fagegaltier et al., 2014*; *Mohn et al., 2014*; *Quénerch'du et al., 2016*; *Andersen et al., 2017*; *Parhad et al., 2017*). We indeed detected satDNA-derived small RNAs in testes and ovaries (*Figure 2—figure supplement 1A, B*). Our results suggest that the majority of these satDNA-derived small RNAs are piRNAs. First, these small RNAs are abundant in testes and ovaries, and their size distribution is typical for piRNA populations: an average of 90% of the RNAs range from 23 nt to 28 nt, with a peak at 24–26 nt in *D. melanogaster* (*Brennecke et al., 2007*; *Figure 2A* for *Rsp* and *Figure 2—figure supplement 1C* for *1.688*). Second, the satDNA-derived small RNAs bear a signature of the piRNA-guided RNA cleavage process called the ping-pong cycle. piRNAs amplified through ping-pong have a 10 nt overlap of antisense-sense piRNAs with a preference of uridine at the 5′ end (1U) or adenosine at nucleotide position 10 (10A) (*Brennecke et al., 2007*; *Gunawardane et al., 2007*). Our analysis of the ovary small RNA-seq data (*Mohn et al., 2014*; *Andersen et al., 2017*; *Parhad et al., 2017*) confirms a ping-pong signature for satDNA-derived small RNAs: Z-score = 4.55 for *Rsp* and 6.85 for *1.688* satellite (*Figure 2—figure supplement 1E, G*) and ~60–80% have 1U/10A (*Figure 2—figure supplement 1D, F*). Third, satDNA-derived small RNAs are bound by the PIWI proteins, as expected for piRNAs. Our reanalysis of published Piwi, Aubergine (Aub), and Argonaute3 (Ago3) RIP-seq data from ovaries (*Brennecke et al., 2007*; *Mohn et al., 2015*; *Sato et al., 2015*) shows that *Rsp* and *1.688* RNAs interact with each of these proteins (*Supplementary file 5*). For example, ~0.9% and 0.1% of Piwi-bound RNAs map to *1.688* and *Rsp*, respectively. For comparison, ~2% and 17% Piwi-bound RNAs mapped to the dual-strand piRNA clusters *80F* and *42AB*, respectively. In contrast, only an average of 0.0005% of the reads from Piwi RIP-seq data mapped to miRNAs, which are abundant small RNAs not known to be bound by Piwi. This suggests that the abundance of satellite RNA in the RIP-seq data is not likely due to noise or contamination. Our results from Aub and Ago3 RIP data are similar to Piwi (*Supplementary file 5*; e.g., 3.1% and 0.1% of Aub-bound RNAs map to *1.688* and *Rsp*, respectively; and 1.8% and 0.07% of Ago3-bound RNAs map to *1.688* and *Rsp*, respectively).

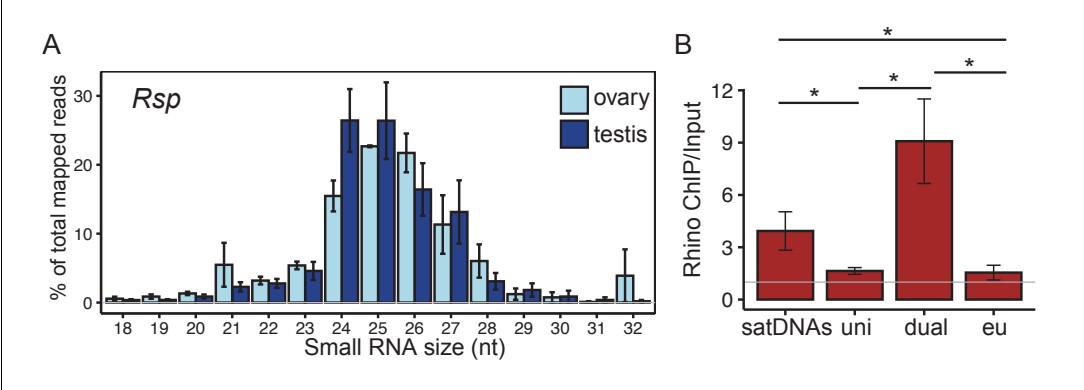

**Figure 2.** Satellite DNAs (satDNAs) produce small RNAs in *D. melanogaster* ovaries. (**A**) Size distribution of *Rsp* small RNAs in testes and ovaries (*1.688* distribution is in ***Figure 2—figure supplement 1C***). Source data in ***Figure 2—source data 1***. (**B**) Rhino ChIP-seq result from ovaries showing the enrichment scores for satDNAs, uni-strand (uni) piRNA clusters, dual-strand (dual), and euchromatin (eu). The enrichment scores for each satDNA and piRNA cluster are shown in ***Figure 2—figure supplement 5A***. p-values are estimated by pairwise t-tests with FDR correction (***Benjamini and Hochberg, 1995***). * adjusted p-value<0.05. Source data in ***Figure 2—source data 2***.

The online version of this article includes the following source data and figure supplement(s) for figure 2:

**Source data 1.** Size distribution of small RNAs for *Rsp* and *1.688* in testes and ovaries.
**Source data 2.** Rhino/H3K9me3 ChIP-seq enrichment scores for *Rsp*, *1.688* heterochromatic loci, piRNA clusters, and euchromatin.
**Source data 3.** Rhino ChIP-seq enrichment scores for all repeats in the genome.
**Figure supplement 1.** Satellite-derived RNAs are mainly processed into piRNAs in the germline.
**Figure supplement 1—source data 1.** Small RNA levels (RPM) in ovary, testis, head, and body (minus head) for *1.688* and *Rsp*.
**Figure supplement 2.** Non-uniform distribution of piRNA reads along satellite DNA (satDNA) consensus sequences.
**Figure supplement 3.** Non-uniform distribution of piRNA reads along the germline-dominant transposable element (TE) consensus sequences.
**Figure supplement 4.** Satellite DNAs (satDNAs) are transcribed from both strands.
**Figure supplement 4—source data 1.** Percentages of reads mapped to plus and minus strands at all genomic copies of *Rsp* or *1.688* satellites.
**Figure supplement 5.** ChIP-seq result shows the chromatin state of satellite DNAs (satDNAs), uni-strand piRNA clusters, dual-strand piRNA clusters, and euchromatin (eu).

Together, these results indicate that satDNA-derived transcripts are processed into piRNAs in the female germline.

We examined the piRNA distribution along individual repeat units for *Rsp* and two subfamilies of *1.688* (*359-bp* and *260-bp*) by blasting the corresponding sequencing reads to each consensus sequence. We find that the distribution of piRNA read depth is not uniform along the length of single monomers (*359-bp* and *260-bp*) or dimers (*Rsp*) of these satDNA repeats (***Figure 2—figure supplement 2A, B***). This pattern could arise if these piRNAs derive from repeat fragments overrepresented in the genome. However, when we look at the alignment depth of all genomic repeat variants, it is more uniform across the monomers/dimers than the piRNA pileup (***Figure 2—figure supplement 2C***). We observe similar non-uniform patterns of piRNA pileup over germline-dominant TEs (e.g., *invader6*, *mdg3*, and *Het-A*; ***Figure 2—figure supplement 3***), suggesting that these uneven distributions may arise from piRNA processing. The piRNA read pileup pattern also differs between ovaries and testes (***Figure 2—figure supplement 2A, B***), suggesting that there may be differences in transcription machinery, precursor production, or precursor processing between these tissues.

## SatDNA transcription resembles dual-strand piRNA clusters

*D. melanogaster* ovarian piRNAs originate primarily from uni- or dual-strand piRNA clusters. To determine which pathway controls the expression of satDNA-derived piRNA precursors, we asked whether transcripts come from one or both strands. We mapped total RNAseq reads from ovary and testis to the genome assembly. Collectively, for all genomic copies of *Rsp* or *1.688* satDNA (all subfamilies), we find a nearly 1:1 ratio of reads mapping to the plus and minus strands (***Figure 2—figure supplement 4A***; all mapped and uniquely mapped reads). However, the highly repetitive nature of satDNAs makes confidently assigning satellite-derived reads to a genomic location difficult. We

therefore take advantage of our assemblies for two representative satDNA loci: the major *Rsp* locus on chromosome 2R and the *260-bp* locus, a subfamily of *1.688*, on chromosome 2L (*Khost et al., 2017*). For these two loci, we confirm that reads map uniquely to both strands of the contigs (*Figure 2—figure supplement 4B*). Together, these results suggest that satDNAs are transcribed from both strands, similar to dual-strand piRNA clusters.

Dual-strand piRNA clusters are associated with the heterochromatin binding protein Rhi (*Klattenhoff et al., 2009*; *Zhang et al., 2014*). We therefore reanalyzed publicly available ChIP-seq datasets from ovaries (*Mohn et al., 2014*; *Zhang et al., 2014*; *Parhad et al., 2017*) to determine if satDNA regions are also Rhi-associated. Our results for piRNA clusters are consistent with previous studies (*Klattenhoff et al., 2009*; *Mohn et al., 2014*; *Andersen et al., 2017*): the dual-strand piRNA clusters have higher Rhi enrichment (mean enrichment ChIP/Input $E_{dual}$ = 9.08) compared to uni-strand piRNA clusters ($E_{uni}$ = 1.69; pairwise t-test with Benjamini–Hochberg; *Benjamini and Hochberg, 1995* adjusted p-value $P_{adj}$=0.01) and euchromatic genes ($E_{euch}$ = 1.55; $P_{adj}$=0.01). We found that complex satDNAs are in the top 30% of all repeats enriched in Rhi (full data in *Figure 2—source data 3*). The level of Rhi enrichment for satDNAs ($E_{sat}$ = 4.70) is intermediate between the highly enriched dual-strand piRNA clusters ($P_{adj}$=0.1) and the minimally Rhi enriched uni-strand piRNA clusters ($P_{adj}$=0.01) or euchromatin ($P_{adj}$=0.01 *Figure 2B* and *Figure 2—figure supplement 5A*). Unlike the uneven distribution of piRNAs along satellite monomers/dimers (*Figure 2—figure supplement 2A, B*), the distribution of Rhi ChIP-seq reads (*Figure 2—figure supplement 2D*) is similar to the alignment depth of genomic repeats (*Figure 2—figure supplement 2C*). This suggests that Rhi localizes to the large satDNA genomic loci rather than a subset of smaller clusters or repeats across the genome (e.g., the 12 copies of *Rsp* inside an intron of *Ago3* on chromosome 3L; *Figure 2—figure supplement 2C*) or in potentially unannotated piRNA clusters.

## SatDNA transcription is regulated by RDC complex and Moon

Because we find that satDNAs generate piRNAs in the female germline and their chromatin is associated with Rhi, we asked if the same transcription and RNA processing machinery are used by both satDNAs and dual-strand piRNA clusters. We used publicly available small RNA-seq datasets generated from mutants of genes involved in the heterochromatin-dependent transcription initiation of dual-strand piRNA clusters: Rhi, Cuff, Del (RDC), and Moon (*Klattenhoff et al., 2009*; *Pane et al., 2011*; *Czech et al., 2013*; *Le Thomas et al., 2014*; *Mohn et al., 2014*; *Andersen et al., 2017*; *Parhad et al., 2017*). We normalized piRNA abundance to the number of reads mapped to either miRNAs (*Figure 3A*) or the uni-strand *flamenco* cluster (*Figure 3—figure supplement 1*), neither of which should be affected by mutations in the RDC pathway.

Our analysis of the known piRNA clusters agrees with published results: the dual-strand piRNA clusters *42AB* and *80F* are Rhi- and Moon-dependent, and *38C1/2* is Rhi-dependent but not Moon-dependent. The uni-strand piRNA clusters *20A* and *flamenco* are not dependent on either protein (*Klattenhoff et al., 2009*; *Pane et al., 2011*; *Mohn et al., 2014*; *Andersen et al., 2017*). We find that the pools of complex satDNA-derived piRNAs are also reduced in RDC and Moon mutants (*Figure 3A*, *Figure 3—figure supplement 1*). In *rhi* mutants, *Rsp* piRNA abundance is 0.2–6.3% their levels in wild-type datasets. Similarly, piRNA abundance for *1.688* is 1.4–7.8% their levels in wild-type datasets (complete list of log2 fold change for satDNAs in *Supplementary files 6* and *7*). The reduction in satDNA piRNA level is robust to normalization method (miRNA in *Figure 3A*; *flamenco* cluster in *Figure 3—figure supplement 1*). While the expression of simple satellite repeats like AAGAG was not decreased in these mutants (*Supplementary file 6* and *Supplementary file 7*), the low abundance of AAGAG reads (the number of reads mapping to AAGAG are only ~0.5% of *Rsp* and ~0.03% of *1.688*) and known sources of bias for simple repeats (e.g., PCR bias in RNA-seq library preparation; *Wei et al., 2018*) points to the need for different approaches to verify this finding. Overall, our results indicate that piRNA production from complex satDNAs is regulated by the heterochromatin-dependent transcription machinery associated with dual-strand piRNA clusters.

To further examine how the RDC complex and Moon affect complex satDNA transcription, we reanalyzed total RNA-seq data of the corresponding mutants (*Mohn et al., 2014*; *Andersen et al., 2017*). RDC and Moon mutants affect piRNA precursor transcription at the dual-strand piRNA clusters *42AB* and *80F* (*Mohn et al., 2014*; *Andersen et al., 2017*). Consistent with published reports, we detected decreases in steady-state long RNA transcript levels at dual-strand piRNA clusters (*Figure 3—figure supplement 2*). However, we did not observe a significant decrease in steady-state

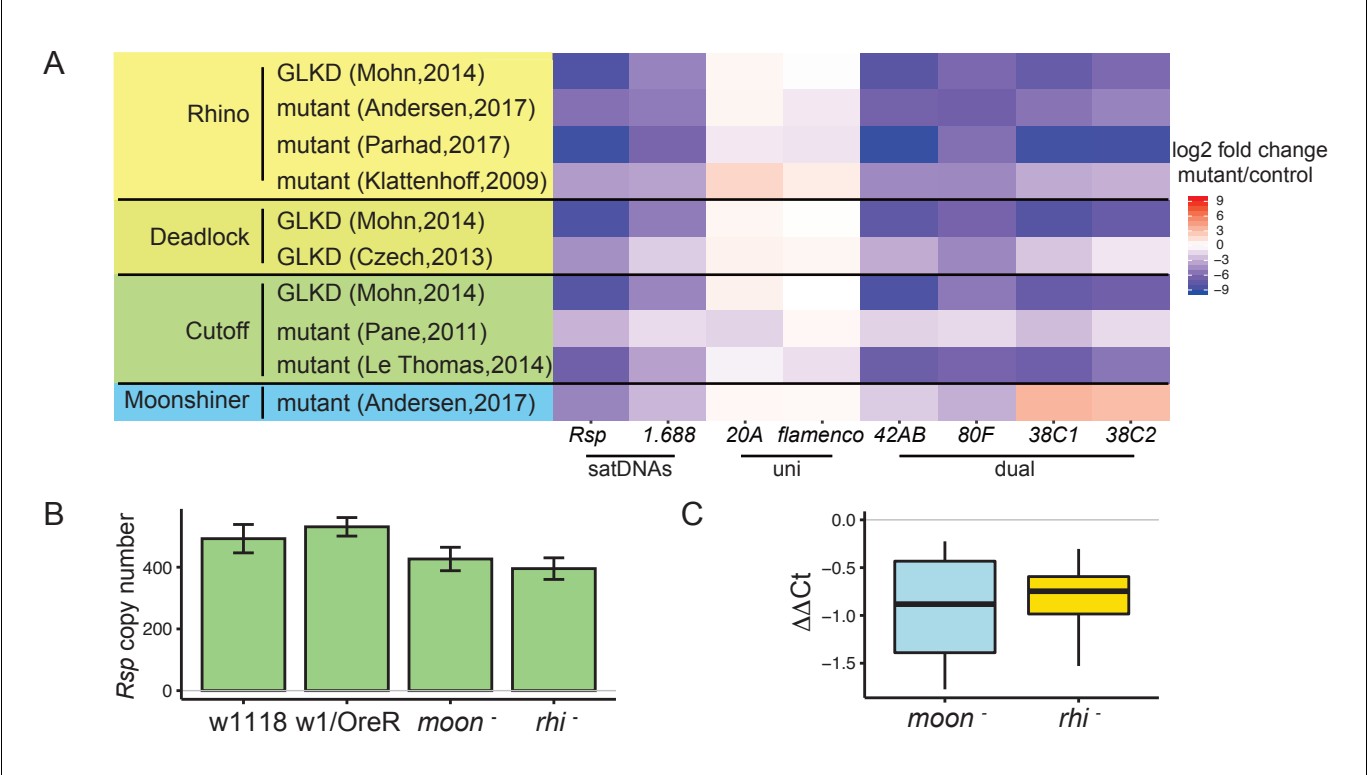

**Figure 3.** Satellite DNA (satDNA) loci are regulated by the heterochromatin-dependent transcription machinery in *Drosophila* ovaries. (**A**) Heatmap showing the quantification of changes in piRNA abundance in small RNA-seq data from mutants of *rhino*, *cutoff*, *deadlock*, and *moonshiner* compared to controls for satDNAs and piRNA clusters, normalized by miRNA level. GLKD: germline knockdown. Complete list of log2 fold changes in *Supplementary file 6*. (**B**) qPCR estimate of *Rsp* copy number in wild types and mutants. (**C**) qRT-PCR estimate of *Rsp* transcript level in mutants compared to wild types. ΔΔCt = ΔCt(wild type) – ΔCt(mutant), a negative value indicates lower expression in mutant. Student's t-test, p-value=0.077, 0.048. Source data in *Figure 3—source data 1*.

The online version of this article includes the following source data and figure supplement(s) for figure 3:

**Source data 1.** *Rsp* copy number and expression level estimated from qPCR and q-RT-PCR.

**Figure supplement 1.** Satellite DNA (satDNA) loci are regulated by the heterochromatin-dependent transcription machinery in *Drosophila* ovaries.

**Figure supplement 2.** Rhino, Deadlock, and Cutoff (RDC) and *moon* mutants affect piRNA precursor transcription at piRNA clusters.

**Figure supplement 2—source data 1.** Log2 fold changes of total RNA abundance for piRNA clusters from Rhino, Deadlock, and Cutoff (RDC) and Moon mutants.

**Figure supplement 3.** Satellite DNA (satDNA) piRNA production is affected in mutants of pathways involving piRNA precursor export, primary piRNA biogenesis, and the ping-pong cycle.

**Figure supplement 3—source data 1.** Log2 fold changes of small RNA abundance for satellite DNAs (satDNAs) and piRNA clusters from mutants of proteins in the primary piRNA pathway, pathway for piRNA precursor export from the nucleus and ping-pong pathway.

long RNA transcript levels for satDNAs (*Supplementary file 8*). To confirm this finding, we performed qRT-PCR using total RNA from ovaries of *rhino* (*rhi-*) and *moonshiner* (*moon⁻*) mutants (*Andersen et al., 2017*). After controlling for genomic repeat copy number with qPCR (*Figure 3B*), *Rsp* expression level is lower, but not significantly so in *rhi* and *moon* mutants compared to wild type (p-value=0.048 and 0.077; *Figure 3C*). Because satDNAs have generally low expression levels (*Rsp* and *1.688* total RNA levels are ~3% and ~25%, respectively, of both *42AB* and *80F*), we may have insufficient power to detect decreased expression in the mutants. It is also possible that the signal is masked by non-precursor transcripts. That is, there may be two kinds of transcription at satDNA loci: (1) RDC-regulated transcription that generates non-polyadenylated piRNA precursors and, (2) non-precursor transcription, which is not well characterized and may also largely lack polyadenylation. In this context, it would be difficult to distinguish precursor from non-precursor transcripts derived from satDNA. However, when we reanalyzed the total and poly-A selected RNA-seq data from the *rhi* mutant (*ElMaghraby et al., 2019*), we find that the abundance of poly-A

transcripts (which are likely a subset of non-precursors) is increased for *Rsp* and unchanged for *1.688* (*Supplementary file 9*) relative to wild type. This result suggests that changes in piRNA precursor levels may be masked by the non-precursor levels, similar to reports on piRNA cluster transcription in embryonic *piwi* knockdown ovaries (*Akkouche et al., 2017*). This situation might arise if only a subset of satDNA repeats are RDC-regulated. Alternatively, the proportion of piRNA precursor-to-non-precursor transcripts in these mutants might shift such that the abundance of piRNA precursors decreases but the total RNA level does not.

We also asked if the satDNA-derived piRNA pool is affected in mutants of 12 genes involved in piRNA precursor export from the nucleus, primary piRNA biogenesis, and the ping-pong cycle (*Figure 3—figure supplement 3*; *Czech et al., 2018*; datasets from *Malone et al., 2009*; *Handler et al., 2011*; *Olivieri et al., 2012*; *Preall et al., 2012*; *Zhang et al., 2012*; *Czech et al., 2013*; *Sato et al., 2015*; *Wang et al., 2015*; *Supplementary file 1*). For each of the datasets analyzed, we recapitulate previously reported results for all known piRNA clusters (*Figure 3—figure supplement 3*; *Czech et al., 2018*). Our reanalysis of these data suggests that piRNA production for all complex satDNA is regulated by the primary piRNA pathway (Gasz, Vreteno, Shutdown), UAP56, and the ping-pong pathway (Ago3, Krimper). Some of our reanalysis results varied between datasets from different studies for satDNAs. For example, satDNAs show decreased piRNA levels in one mutant Zucchini dataset (*Olivieri et al., 2012*) but increased levels in an independent Zucchini dataset (*Malone et al., 2009*; *Handler et al., 2011*). While further work is required to determine all of the components involved in processing satDNA transcripts, our results suggest that piRNA production at satDNA loci is regulated by the dual-strand piRNA pathway.

## Heterochromatin establishment at satDNAs requires Piwi

Consistent with their Rhi enrichment, we find that satDNAs are enriched for H3K9me3 in ovaries (*Figure 2—figure supplement 5B*; datasets from *Klenov et al., 2014*; *Le Thomas et al., 2014*; *Mohn et al., 2014*). Piwi plays an important role in establishing H3K9 methylation on euchromatic TEs in ovaries (*Mohn et al., 2014*) and heterochromatin more generally in embryos (*Akkouche et al., 2017*). Transiently knocking down *piwi* expression early in the embryonic germline leads to a general decrease in H3K9me3 in the adult ovary, and a specific decrease in piRNA production and increase in spliced non-precursor transcripts at dual-strand piRNA clusters (*Akkouche et al., 2017*). We therefore reanalyzed H3K9me3 ChIP-seq data from embryonic *piwi* knock down ovaries (*Akkouche et al., 2017*). We detected a decrease of H3K9me3 at satDNAs (*Figure 4A*), suggesting that Piwi is also required for the establishment of heterochromatin at these loci. Consistent with the decrease in H3K9me3, piRNA production from satDNAs is also reduced (with some variation among replicates observed for *Rsp*; *Figure 4B*); and satDNA total RNA levels are increased (*Figure 4—figure supplement 1*), similar to dual-strand piRNA clusters (*Akkouche et al., 2017*). However, it is again difficult to distinguish between satDNA precursor and non-precursor RNAs.

While Piwi is important for heterochromatin establishment at piRNA clusters, it appears to be dispensable for heterochromatin maintenance (*Czech et al., 2018*). Depleting Piwi in the nucleus with *piwi* mutants lacking a nuclear localization signal (NLS; *Klenov et al., 2014*), or knocking down germline *piwi* (*Le Thomas et al., 2013*; *Mohn et al., 2014*) affects H3K9me3 level on a subset of active transposons, but not on piRNA clusters (*Klenov et al., 2014*; *Mohn et al., 2014*). Similar to piRNA clusters, our reanalysis of these data shows that the level of H3K9me3 on satDNAs is largely unchanged in the knockdown or mutant ovaries (with some variation observed among datasets; *Supplementary file 10*). These analyses suggest a role for Piwi in establishing, but not maintaining, heterochromatin at satDNAs in early embryos, which is important for producing piRNAs later in adult ovaries.

## Conclusions

piRNA pathways are primarily studied for their conserved role in protecting genome integrity by repressing TE activity in different organisms (*Aravin et al., 2006*; *Girard et al., 2006*; *Grivna et al., 2006*; *Lau et al., 2006*; *Brennecke et al., 2007*; *Houwing et al., 2007*; reviewed in *Parhad and Theurkauf, 2019*). However, our findings support a more general role for these pathways. Here we show that transcription from satDNAs is regulated by the heterochromatin-dependent RDC

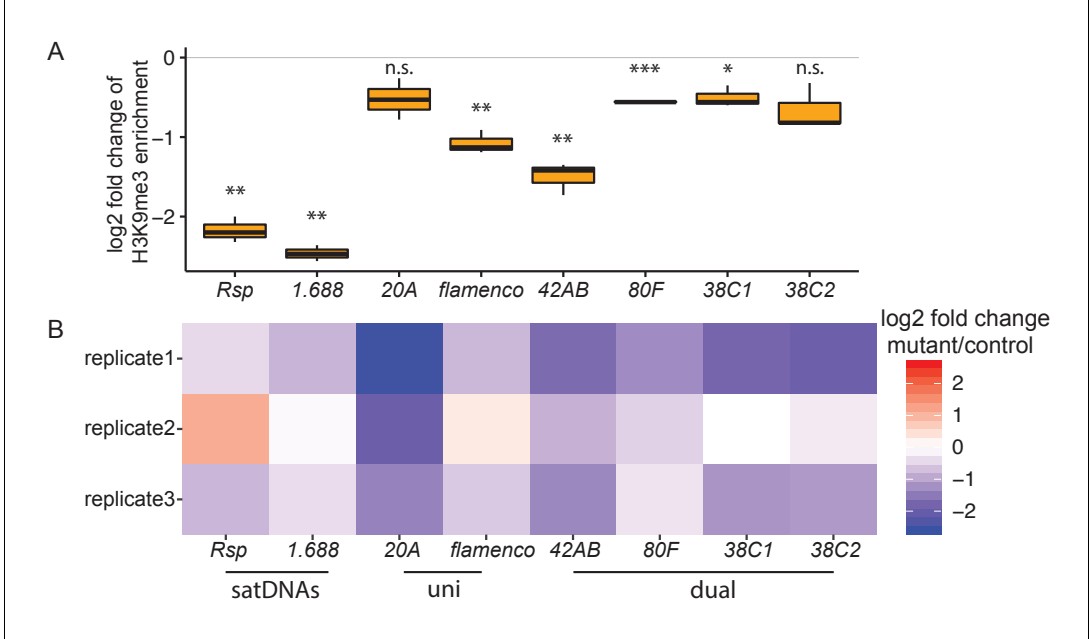

**Figure 4.** Heterochromatin establishment disrupted at satellite DNAs (satDNAs) in *piwi* embryonic knockdown ovaries. (**A**) Log2 fold change of H3K9me3 ChIP/input enrichment shows satDNA H3K9me3 levels decrease in *piwi* embryonic knockdown ovaries compared to control. Source data in *Figure 4—source data 1*. p-values are estimated by one-sample t-test (mu = 0) with FDR corrections (*Benjamini and Hochberg, 1995*). * adjusted p-value<0.05, ** adjusted p-value<0.01, *** adjusted p-value<0.001. (**B**) Log2 fold change of small RNA abundance shows satDNA small RNA levels decrease compared to controls, with variation observed for replicate2. Small RNA abundance is normalized to the number of reads mapped to miRNAs. Source data in *Figure 4—source data 2*.

The online version of this article includes the following source data and figure supplement(s) for figure 4:

**Source data 1.** Log2 fold change of H3K9me3 ChIP/input enrichment for satellite DNAs (satDNAs) and piRNA clusters in *piwi* embryonic knockdown ovaries.

**Source data 2.** Log2 fold change of small RNA abundance for satellite DNAs (satDNAs) and piRNA cluters in *piwi* embryonic knockdown ovaries.

**Figure supplement 1.** Log2 fold change of total RNA abundance shows satellite DNA (satDNA) long RNA levels increase in *piwi* embryonic knockdown ovaries compared to control.

**Figure supplement 1—source data 1.** Log2 fold change of total RNA abundance for satellite DNA (satDNA) and piRNA clusters in *piwi* embryonic knockdown ovaries.

machinery and Moon in ovaries and these transcripts are processed into piRNAs. Thus, complex satDNA transcription is regulated in a manner similar to dual-strand piRNA clusters in the female germline (*Figure 5*).

Our findings are consistent with a study that detected bidirectional transcription of the *1.688* satDNA family in ovaries (*Usakin et al., 2007*) and a recent analysis of satDNA-derived piRNAs in RDC mutants (*Chen et al., 2020*). Usakin et al. found that *1.688* transcript abundance is elevated in mutants of two piRNA processing genes, *spn-E* and *aub* (*Usakin et al., 2007*), suggesting that *1.688* is targeted by piRNAs, similar to TEs. However, the origins of the *1.688* piRNAs and how the transcription of precursors is regulated were unclear (*Usakin et al., 2007*). Here we provide evidence that most satellite-derived transcripts and small RNAs reported in previous studies (*Aravin et al., 2003*; *Saito et al., 2006*; *Usakin et al., 2007*; *Rošić et al., 2014*; *Chen et al., 2020*; *Chen et al., 2021*) come from the heterochromatin-dependent transcription of the large satDNA blocks. The role of these piRNAs in ovaries remains unknown, and we understand even less about piRNA biogenesis and function in *D. melanogaster* testes, where we also detect satDNA-derived piRNAs. Proportionally, far fewer piRNAs in the male germline are derived from TEs than in the female germline (*Nishida et al., 2007*; *Nagao et al., 2010*; *Quénerch'du et al., 2016*), suggesting roles outside of TE repression. For example, recent studies implicate piRNA pathways in intragenomic conflicts (e.g., male meiotic drive; *Gell and Reenan, 2013*; *Courret et al., 2019*), with satDNAs often at the center of these conflicts.

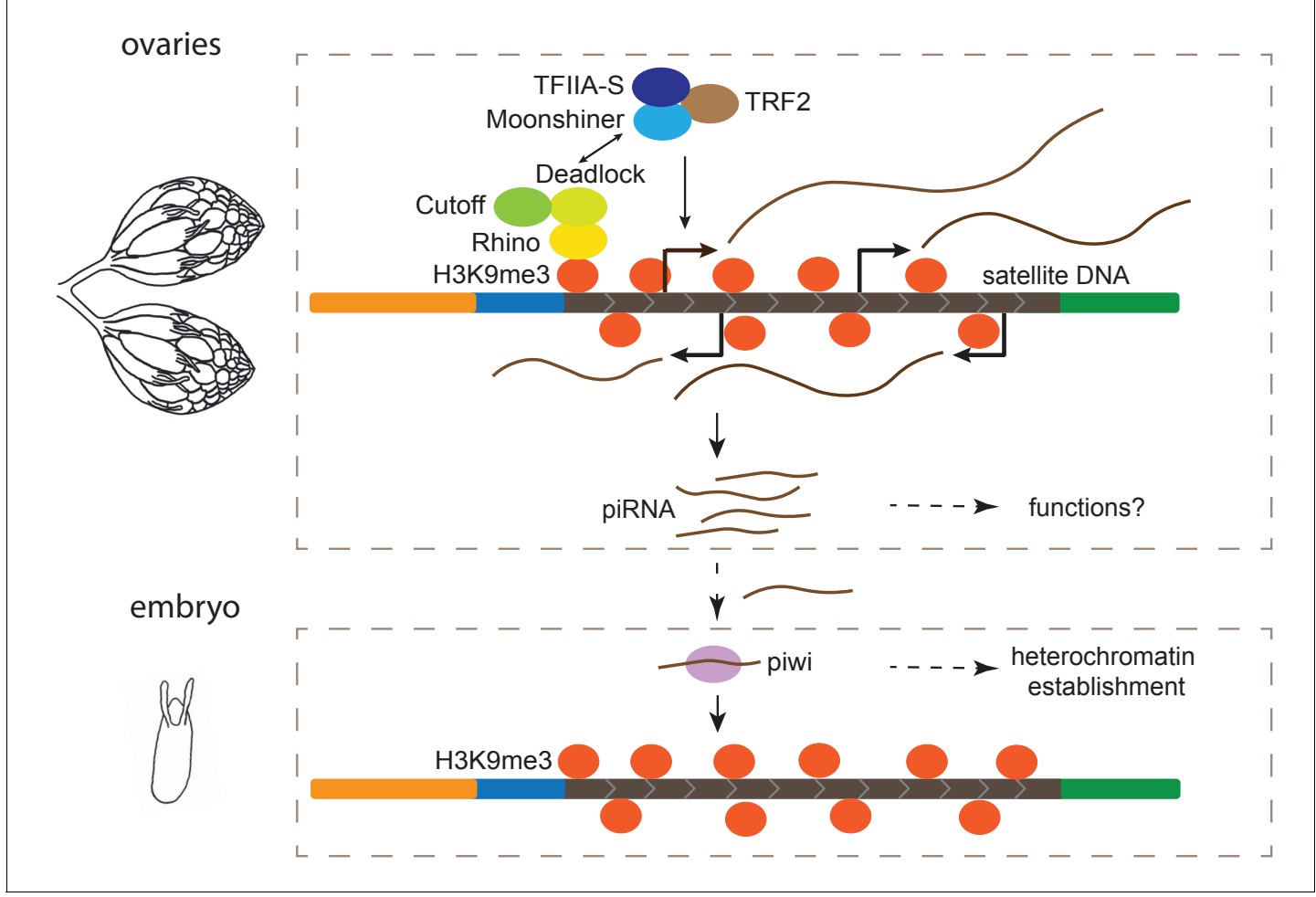

**Figure 5.** Model for maintenance of satellite DNA (satDNA) chromatin in female germline. Complex satDNA transcription is regulated by the heterochromatin-dependent Rhino-Deadlock-Cutoff and Moonshiner machinery, and the long RNA transcripts are processed into piRNAs. While their functions in ovaries are unclear, these piRNAs play roles in the establishment of heterochromatin at their own genomic loci in embryos. This pathway may be important for maintaining genome stability in pericentric heterochromatin, proper nuclear organization, and other unexplored functions.

While it will take more work to understand the role of satDNA-derived transcripts in the germline, we hypothesize that the maternal deposition of these piRNAs contributes to heterochromatin establishment at satDNAs in the early embryo (*Figure 5*). Maternal deposition of Piwi contributes to heterochromatin establishment in the embryo (*Gu and Elgin, 2013*), and Piwi-dependent H3K9me3 deposition at canonical piRNA clusters is important for subsequent piRNA production at piRNA clusters (*Akkouche et al., 2017*). Similar to piRNA clusters, we found evidence that both H3K9me3 chromatin and piRNA production from complex satDNA is reduced when transiently depleting Piwi in the embryos, suggesting a role for the piRNA pathway in heterochromatin establishment at satDNA loci (*Figure 5*). We propose a simple model of self-regulation, where Piwi, guided by satDNA-derived piRNAs, establishes H3K9me3 at satDNA, marking the satDNAs as piRNA production sites later in development (*Figure 5*). While a contributor, Piwi might not be the only factor necessary for heterochromatin establishment in embryos (*Wei et al., 2021a*). And once established, the maintenance of heterochromatin at piRNA clusters and satDNAs is not Piwi-dependent (*Supplementary file 10*; *Klenov et al., 2014*; *Mohn et al., 2014*). Therefore, the piRNA pathway is likely to be one of several factors important for proper packaging and regulation of repeat-rich regions of the genome (*Pal-Bhadra et al., 2004*; *Gu and Elgin, 2013*).

The consequences of disrupting satDNA packaging/regulation are likely to be complicated. The ramifications could be especially serious if a reduction in heterochromatin at satDNA in early

embryos affects heterochromatin in all tissues (reviewed in *Janssen et al., 2018*) and/or if establishing heterochromatin at satDNA loci serves as nucleation points for the larger-scale heterochromatinization of pericentric regions. First, heterochromatic regions form a distinct phase-separated nuclear compartment that contributes to nuclear organization and gene regulation (*Larson et al., 2017*; *Strom et al., 2017*), and chromocenter formation (*Jagannathan et al., 2018*). Unregulated satDNA may disrupt this organization (*Novo et al., 2020*) and lead to cell death (*Jagannathan et al., 2019*). Second, de-repressed satDNA may lead to genome instability (*Peng and Karpen, 2007*) including chromosomal structural rearrangements (reviewed in *Janssen et al., 2018*). In the short term, rearrangements involving satDNA may lead to mitotic defects in the developing embryo as they can affect chromosome segregation (*Ferree and Barbash, 2009*; *Ferree, 2014*). Over longer evolutionary time scales, these rearrangements contribute to variation in satDNA organization between individuals and species, and may cause genetic incompatibilities between closely related species (*Ferree and Barbash, 2009*). SatDNAs are indeed among the most rapidly evolving sequences in genomes (reviewed in *Ferree and Prasad, 2012*; *Plohl et al., 2012*).

Many mysteries remain surrounding the functions of the piRNA pathway outside of its role in controlling TE activity. Our finding that the piRNA pathway regulates satDNA suggests a general role for the piRNA pathway and for maternal satDNA-derived RNAs in remodeling chromatin in the developing embryo. This initial establishment of heterochromatin may be an important step in ensuring genome integrity throughout development and in adult tissues, but this remains an open question. Moving forward, it will be important for piRNA studies to continue to focus on satDNA and how these dynamic compartments of the genome contribute to genome function and stability.

# Materials and methods

## Key resources table

| Reagent type (species) or resource | Designation | Source or reference | Identifiers | Additional information |
|---|---|---|---|---|
| Gene (*Drosophila melanogaster*) | *rhi* | Flybase | Flybase: FBgn0004400 | |
| Gene (*D. melanogaster*) | *moon* | Flybase | Flybase: FBgn0030373 | |
| Gene (*D. melanogaster*) | *del* | Flybase | Flybase: FBgn0086251 | |
| Gene (*D. melanogaster*) | *cuff* | Flybase | Flybase: FBgn0260932 | |
| Gene (*D. melanogaster*) | *piwi* | Flybase | Flybase: FBgn0004872 | |
| Strain, strain background (*D. melanogaster, female and male*) | Iso-1 | Bloomington Drosophila Stock Center (BDSC) | BDSC: 2057; RRID:BDSC_2057 | |
| Strain, strain background (*D. melanogaster, female*) | ZW144 | doi:10.1534/g3.114.015883 *Grenier et al., 2015* | | |
| Strain, strain background (*D. melanogaster, female*) | Ral357 | BDSC | BDSC:25184; RRID:BDSC_25184 | |
| Strain, strain background (*D. melanogaster, female*) | Ral380 | BDSC | BDSC:25190; RRID:BDSC_25190 | |
| Strain, strain background (*D. melanogaster, female*) | lt pk cn bw | *Ganetzky, 1977* | | |
| Strain, strain background (*D. melanogaster, female*) | w[1118] | BDSC | BDSC:5905; RRID:BDSC_5905 | |

*Continued on next page*

*Continued*

| Reagent type (species) or resource | Designation | Source or reference | Identifiers | Additional information |
|---|---|---|---|---|
| Strain, strain background (*D. melanogaster, female*) | w[1] | BDSC | BDSC:2390; RRID:BDSC_2390 | |
| Strain, strain background (*D. melanogaster, female*) | OregonR | BDSC | BDSC:2376; RRID:BDSC_2376 | |
| Genetic reagent (*D. melanogaster*) | *rhi* mutant | Vienna Drosophila Resource Center (VRDC) | VDRC:313487 | |
| Genetic reagent (*D. melanogaster*) | *rhi* mutant | VRDC | VDRC:313488 | |
| Genetic reagent (*D. melanogaster*) | *moon* mutant | VRDC | VDRC:313735 | |
| Genetic reagent (*D. melanogaster*) | *moon* mutant | VRDC | VDRC:313738 | |
| Sequence-based reagent | RPS3 forward | IDT | qPCR primer | AGTTGTACGCCGAGAAGGTG |
| Sequence-based reagent | RPS3 Reverse | IDT | qPCR primer | TGTAGCGGAGCACACCATAG |
| Sequence-based reagent | tRNA forward | IDT | qPCR primer | CTAGCTCAGTCGGTAGAGCATGA |
| Sequence-based reagent | tRNA Reverse | IDT | qPCR primer | CCAACGTGGGGCTCGAAC |
| Sequence-based reagent | *Rsp* forward | IDT | qPCR primer | GGAAAATCACCCATTTTGATCGC |
| Sequence-based reagent | *Rsp* Reverse | IDT | qPCR primer | CCGAATTCAAGTACCAGAC |
| Sequence-based reagent | Probe for *1.688* | IDT | RNA FISH probe | Cy5TTTTCCAAATTTCGGT CATCAAATAATCAT |
| Sequence-based reagent | Probe for *Rsp* | Stellaris | RNA FISH probe | Custom Stellaris FISH probes with 45 sequences listed in ***Supplementary file 11*** |
| Sequence-based reagent | T7_rsp2 | IDT | Northern blot probe synthesis primer | TAATACGACTCACTATAGGG CCGAATTCAAGTACCAGAC |
| Sequence-based reagent | rsp1 | IDT | Northern blot probe synthesis primer | GGAAAATCACCCATTTTGATCGC |
| Sequence-based reagent | *Rsp* primer_F | IDT | Slot blot probe synthesis primer | TAATACGACTCACTATAGGGG AAAATCACCCATTTTGATCGC |
| Sequence-based reagent | *Rsp* primer_R | IDT | Slot blot probe synthesis primer | CCGAATTCAAGTACCAGAC |
| Sequence-based reagent | rp49 primer_F | IDT | Slot blot probe synthesis primer | TAATACGACTCACTATAGGG CAGTAAACGCGGTTCTGCATG |
| Sequence-based reagent | rp49 primer_R | IDT | Slot blot probe synthesis primer | CAGCATACAGGCCCAAGATC |
| Software, algorithm | Bowtie2 | doi:10.1038/nmeth.1923. | RRID:SCR_016368 | |
| Software, algorithm | Bowtie | doi:10.1002/0471250953.bi1107s32. | RRID:SCR_005476 | |
| Software, algorithm | DESeq2 | doi:10.1186/s13059-014-0550-8. | RRID:SCR_015687 | |
| Software, algorithm | piPipes | doi:10.1093/bioinformatics/btu647. | | |
| Software, algorithm | BLAST | NCBI | RRID:SCR_004870 | |
| Software, algorithm | R | R core team | RRID:SCR_001905 | |
| Software, algorithm | Customized Python scripts | This paper | | ***Wei et al., 2021b*** GitHub (https://github.com/LarracuenteLab/Dmelanogaster_satDNA_regulation) |

### *Drosophila* stocks

Iso-1 (RRID:BDSC_2057) was used as the wild-type strain, unless stated otherwise. In the qPCR validation experiment, *rhi* mutants (*rhi⁻*) are transheterozygotes from the Vienna Drosophila Resource Center (VDRC 313487 and 313488) as are the *moonshiner* mutants (*moon⁻*) (VDRC 313735 and 313738) as described in *Andersen et al., 2017*. Based on the origin and genetic background of these mutants, *w^1118^* (RRID:BDSC_5905) or the progeny from OregonR (Ore) (RRID:BDSC_2376) crossed to *w^1^* (RRID:BDSC_2390) were used as the wild-type controls for *rhi⁻* and *moon⁻*. All flies were maintained at 23°C on cornmeal medium.

### Small RNA-seq

6–8-day-old testes were dissected in RNase-free PBS buffer. Total RNA was extracted using *mir*Vana miRNA Isolation Kit (Ambion) with procedures for isolating RNA fractions enriched for small RNAs (<200 nt), then treated with RNase free DNase I (Promega) at 37°C for 1 hr. Library preparation and sequencing were performed by Genomics Research Center at University of Rochester. Briefly, 2S rRNA was depleted (*Wickersheim and Blumenstiel, 2013*), small RNA library was prepared with TruSeq Small RNA Library Prep Kit (Illumina) and sequenced by Illumina platform HiSeq2500 Single-end 50 bp.

### Total RNA-seq

6–8-day-old testes were dissected in RNase-free PBS buffer. Total RNA was extracted using *mir*Vana miRNA Isolation Kit (Ambion) with procedures for isolating RNA fractions enriched for long RNAs (>200 nt), then treated with RNase free DNase I (Promega) at 37°C for 1 hr. Library preparation and sequencing were performed by Genomics Research Center at University of Rochester. Briefly, rRNA was removed and total RNA library was prepared with TruSeq Stranded Total RNA Library Prep Human/Mouse/Rat (Illumina) and sequenced by Illumina platform HiSeq2500 Paired-end 125 bp.

### SatDNA analysis

Reads were mapped to the heterochromatin-enriched genome assembly (*Chang and Larracuente, 2019*) and counted based on their annotations (e.g., *Rsp* or *1.688*). Due to the highly repetitive nature of satDNAs, around 80% of total RNA-seq and 99% of small RNA-seq reads that are mapped to satDNA regions are not uniquely assigned; discarding these multiple mapped reads would result in loss of statistical power in the satDNA analysis. To deal with this, multiple mapped reads were randomly assigned to one of their multiple best mapping locations, unless stated otherwise. Reads were then counted based on the annotations of their assigned mapping locations. Because there is high-sequence similarity among the *1.688* subfamily repeats (*260-bp*, *359-bp*, *353-bp*, *356-bp*), all *1.688* subfamilies were combined, unless stated otherwise. A similar approach was used in our analysis of piRNA clusters, except that only uniquely mapped reads were counted so that the published results could serve as controls for our method. Additional details specific to small RNA-seq, RNA-seq, ChIP-seq, and RIP-seq analyses are given below.

### RNA-seq analysis

All total RNA-seq datasets reanalyzed in our study are listed in *Supplementary file 1*. Total RNA-seq reads were trimmed for adaptors and then mapped to the genome using Bowtie2 (RRID:SCR_016368) (*Langmead and Salzberg, 2012*). A customized Python script was used to count reads that mapped to each repeat feature or piRNA cluster, and RPM values were reported by normalizing raw counts to 1,000,000 total mapped reads (*Wei et al., 2021b* https://github.com/LarracuenteLab/Dmelanogaster_satDNA_regulation; *Wei, 2020*, htseq_bam_count_proportional.py; *Wei et al., 2021b*). For the *1.688* subfamilies, all subfamilies were combined into one *1.688* category, although analyzing each by subfamily (e.g., *353-bp*, *356-bp*, *359-bp*, *260-bp*) does not change our conclusions (https://github.com/LarracuenteLab/Dmelanogaster_satDNA_regulation) (*Wei, 2020*; *Wei et al., 2021b*). For results shown in *Supplementary file 8*, DESeq2 (RRID:SCR_015687) (*Love et al., 2014*) was used to perform differential expression analysis of the raw counts with combined data from different studies (*Mohn et al., 2014*; *Andersen et al., 2017*), with experimental condition and associated study as covariates. This analysis method is conservative and leads to smaller log2 fold changes than published results of piRNA clusters. For comparison with the published results, a similar

approach was used to analyze piRNA clusters (*Mohn et al., 2014*; *Andersen et al., 2017*). Briefly, quantification of reads mapping to 1 kb windows inside each piRNA cluster was estimated using a customized Python script (https://github.com/LarracuenteLab/Dmelanogaster_satDNA_regulation; *Wei, 2020*;, htseq_bam_count_proportional.py; *Wei et al., 2021b*), and subsequent differential expression analysis between mutants and wildtype was done using DESeq2 (RRID:SCR_015687) (*Love et al., 2014*; results shown in *Figure 3—figure supplement 2*).

## Small RNA-seq analysis

All small RNA-seq datasets reanalyzed in our study are listed in *Supplementary file 1*. Small RNA-seq reads were trimmed for adaptors, then mapped to the genome using Bowtie (RRID:SCR_005476) (*Langmead, 2010*). A customized Python script (https://github.com/LarracuenteLab/Dmelanogaster_satDNA_regulation; *Wei, 2020*, htseq_bam_count_proportional.py; *Wei et al., 2021b*) was used to count reads that mapped to each repeat feature or piRNA cluster. To control for differences in small RNA abundance and compare across samples, raw counts were then normalized to the number of reads that mapped to either miRNAs or the *flamenco* piRNA cluster. The difference in expression was represented by the log2 fold changes of these normalized counts in mutants compared to wild type (i.e., $\log2(count_{mutant}/count_{WT})$) for each repeat and piRNA cluster. We further analyzed the size distribution and relative nucleotide bias at positions along each satDNA by extracting reads mapped to the satDNA of interest using a customized Python script (https://github.com/LarracuenteLab/Dmelanogaster_satDNA_regulation; *Wei, 2020*, extract_sequence_by_feature_gff.py; *Wei et al., 2021b*). The 10nt overlap Z-score of piRNAs mapped to each satDNA was calculated using piPipes (*Han et al., 2015*). To determine which parts of these repeats are represented in piRNA or ChIP datasets, the read pileup patterns along the consensus sequence of a satDNA were examined (e.g., *Figure 2—figure supplement 2*). Reads (ChIP or piRNA) mapping to a particular satDNA or genomic satDNA variant (as a control) were BLAST-ed to the consensus dimer (for *1.688* satellite) or trimer (for *Rsp* because it has left and right consensus sequences), and then coordinates were converted along a dimer/trimer to coordinates along a monomer/dimer consensus sequence. All plots were made in R (*R Development Core Team, 2017*).

## ChIP/RIP-seq analysis

All total ChIP-seq and RIP-seq datasets reanalyzed in our study are listed in *Supplementary file 1*. ChIP-seq and RIP-seq reads were trimmed for adaptors and mapped to the genome using Bowtie2 (RRID:SCR_016368) (*Langmead and Salzberg, 2012*). A customized Python script (https://github.com/LarracuenteLab/Dmelanogaster_satDNA_regulation; *Wei, 2020*, htseq_bam_count_proportional.py; *Wei et al., 2021b*) was used to count reads that mapped to each repeat feature or piRNA cluster. Raw counts were normalized to 1,000,000 total mapped reads.

For the ChIP-seq results, enrichment scores of each repeat and piRNA cluster were reported by comparing the ChIP sample with the antibody of interest to its no-antibody input control sample. For ChIP-seq analyses, consider satDNA as discrete loci rather than repeat unit types is appropriate because some loci are composed of several repeat types. To examine the large blocks of heterochromatic satDNA chromatin for the Rhi and H3K9me3 ChIP-seq analyses, euchromatic *1.688* satDNAs were excluded and only reads that map uniquely to satDNA loci were analyzed. Heterochromatic satDNA loci were defined as discrete loci on chromosomes: 2L (2L_2: 402701–460225; the *260-bp* locus), 3L (3L_3: 46695–106272; primarily *353-bp* and *356-bp* repeats), and the unmapped contigs (Contig101 and Contig9; *353-bp*, *356-bp*, and *359-bp* repeats). Our conclusions do not change when we look at all reads (not just uniquely mapped; https://github.com/LarracuenteLab/Dmelanogaster_satDNA_regulation; *Wei, 2020*; *Wei et al., 2021b*). These analyses were repeated by combining all *1.688* subfamilies into a single category, and each subfamily was analyzed separately (e.g., all *353-bp* repeats combined) but the conclusions do not change (https://github.com/LarracuenteLab/Dmelanogaster_satDNA_regulation; *Wei, 2020*; *Wei et al., 2021b*). Euchromatic controls are included for the Rhi and H3K9me3 ChIP-seq analyses. Here, the euchromatic control corresponds to the median enrichment score for protein coding genes that are 5 Mb distal from heterochromatin boundaries (*Riddle et al., 2011*) and piRNA clusters.

For the RIP-seq analyses, reported was the percentage of reads mapped to each repeat and piRNA cluster with miRNAs as the negative control. For the *1.688* subfamilies, all subfamilies were

combined into one *1.688* category, although analyzing each by subfamily (e.g., *353-bp*, *356-bp*, *359-bp*, *260-bp*) does not change the conclusions (https://github.com/LarracuenteLab/Dmelanogaster_satDNA_regulation; *Wei, 2020*; *Wei et al., 2021b*).

## RNA FISH

A Cy5-labeled oligo probe (5′-Cy5TTTTCCAAATTTCGGTCATCAAATAATCAT-3′) previously described in *Ferree and Barbash, 2009* was used to detect *1.688* transcripts from all subfamilies except *260-bp* on chromosome *2L*. Custom Stellaris FISH probes were designed for *Rsp* (*Supplementary file 11*), and RNA FISH was performed following the manufacturer's instructions (Biosearch Technologies, Inc). 3–6-day-old ovaries and testes were dissected in RNase-free PBS buffer, fixed with 4% paraformaldehyde in PBS buffer at room temperature for 30 min, and then washed twice with PBS for 5 min. To permeabilize, tissues were kept in RNase free 70% ethanol at 4°C overnight. The ethanol was aspirated, and samples washed with Stellaris wash buffer on a nutating mixer for 3 min and kept still for 2 min at room temperature. Hybridization was then performed with each probe in Stellaris hybridization buffer in a humidity chamber at 37°C overnight. The working concentration was 100 nM for the oligo probe and 125 nM for the Stellaris probes. From this point, samples were kept in the dark. The samples were washed with Stellaris wash buffer twice at 37°C for 30 min each. Samples were then transferred to mounting medium containing DAPI and imaged with Leica SP5 laser scanning confocal microscope.

For RNaseA controls, after fixation and permeation, tissues were treated with RNase A (100 μg/ml) in RNase digestion buffer (5 mM EDTA, 300 mM NaCl, 10 mM Tris-HCl pH 7.5, Cold Spring Harbor Protocols, http://cshprotocols.cshlp.org/content/2013/3/pdb.rec074146.full) at 37°C for 1 hr and washed three times with Stellaris wash buffer at room temperature for 10 min before hybridization.

For RNase H controls, after probe hybridization and washing, tissues were treated with 1.5 μl RNase H (5000 units/ml; New England Biolabs) in 50 μl final volume in 1X RNAse H buffer at 37°C for 2 hr and washed three times with Stellaris wash buffer at room temperature for 10 min before mounting and imaging.

## qPCR

For genomic DNA qPCR, 3–8-day-old flies were mashed with pipette tips for 5–10 s and incubated in buffer (10 mM Tris-Cl pH 8.2, 1 mM EDTA, 25 mM NaCl, 200 μg/ml Proteinase K) at 37°C for 30 min (*Gloor and Engels, 1992*). To extract nucleic acids, a mixture of phenol/Sevag (1:1) of equal volume was added, and the samples vortexed for 45–60 s and then centrifuged for 3–5 min. The aqueous top layers were saved, an equal volume of Sevag added, and the samples vortexed for 30 s then centrifuged for 1 min. The aqueous top layers were saved and a second Sevag extraction performed. Diluted nucleic acid samples (concentration of 0.04 ng/μl) were used for qPCR to determine the repeat copy numbers in the genome. Repeat copy numbers are normalized to the tRNA:Lys-CTT copy numbers.

For RNA qRT-PCR, 3–6-day-old ovaries were dissected in RNase-free PBS buffer, and total RNA was extracted using the *mir*Vana miRNA Isolation Kit (Ambion). RNA samples were treated with RNase free DNase I (Promega) at 37°C for 1 hr. The RNA samples were reverse transcribed using random hexamer primers and M-MuLV Reverse Transcriptase (New England Biolabs) and the resulting cDNA subjected to qPCR. To exclude the possibility of DNA signal in qRT-PCR experiments, controls with no Reverse Transcriptase enzyme were used for all samples in the reverse transcription step. Expression levels were normalized to ribosomal protein S3 (RPS3) expression. To detect the transcript abundance difference between wild-type and mutant, ΔΔCT was calculated (*Livak and Schmittgen, 2001*).

The replicate number for genomic DNA qPCR is 2–4 and for RNA qRT-PCR is 4–6. The sequences of primers used are: *Rsp* (forward: GGAAAATCACCCATTTTGATCGC, reverse: CCGAATTCAAGTACCAGAC); tRNA (forward: CTAGCTCAGTCGGTAGAGCATGA, reverse: CCAACGTGGGGGCTCGAAC); RPS3 (forward: AGTTGTACGCCGAGAAGGTG, reverse: TGTAGCGGAGCACACCATAG).

## Northern blot analysis

### Isolation of total RNA and RNase controls

Stocks of *D. melanogaster* were chosen, which represented a range of *Rsp* repeat copy numbers; flies were collected (0–20 hr old) and aged for 6 days. Ovaries were dissected from approximately 20 females (i.e., 6.0–6.8 days old) from each stock, and total nucleic acid isolated using a standard phenol/Sevag procedure (*Khost et al., 2017*). Total nucleic acid was then treated with DNase I as recommended (20 units; Promega), re-extracted with phenol/Sevag, and ethanol precipitated. Total RNA was resuspended in distilled water. The integrity of the RNA was checked on 1% agarose gels, and the concentration estimated by an optical density at 260 nm.

For RNase controls, 10 μg of total RNA was resuspended in 50 mM NaCl, 5 mM EDTA, 10 mM Tris pH 7.5, 100 μg/ml RNaseA, and incubated at 37°C for 30 min. Samples were phenol/Sevag extracted, 10 μg of ytRNA added as carrier, and ethanol precipitated.

### Northern blot analysis

Total RNA (10 μg)/RNase controls were suspended in 1× MOPS (0.04 M morpholinepropanesulfonic acid [MOPS] pH 7.0, 0.01 M Na acetate, 0.001 M EDTA), 2.2 M formaldehyde, 50% formamide. The RNA was then heated at 65°C for 15 min, placed on ice, and one-tenth volume loading buffer (1× MOPS, 50% formamide, 2.2 M formaldehyde, 4% Ficoll, 0.25% bromophenol blue) added. RNAs were separated on a 1% agarose gel containing 0.5 M formaldehyde/1× MOPS at 40 V for 3 hr. Standard RNA lanes were cut from the gel and stained with ethidium bromide to monitor electrophoresis. Gels were washed for 25 min in sterile water (with four changes). RNA was transferred to GeneScreen Plus nylon membrane (prewet in 10× SSC) by capillary action using 10× SSC. After transfer, the nylon membrane was rinsed in 2× SSC, UV crosslinked, and then baked for 2 hr under vacuum at 80°C. The membrane was prehybridized in 2× SSC, 5× Denhardt's solution, 1% sodium dodecyl sulfate (SDS), 10% polyethylene glycol (PEG- molecular weight, 8,000), 25 mM sodium phosphate (pH 7.2), 0.1% sodium pyrophosphate, and 50% formamide for 3 hr at 55°C. Hybridizations were done overnight at 55°C in the same buffer containing a biotinylated RNA probe (see slot blot; primers: T7_rsp2 5′-TAATACGACTCACTATAGGGCCGAATTCAAGTACCAGAC-3′ and rsp1 5′-GGAAAATCACCCATTTTGATCGC-3′). The hybridized membranes were washed in 1 M sodium phosphate pH 6.8, 0.5 M EDTA, 5% SDS (2×, 10 min each) at 60°C and then at 1 M sodium phosphate pH 6.8, 0.5M EDTA, 1% SDS (3×, 10 min each) at 65°C. The washed membranes were then processed as recommended for the Chemiluminescent Nucleic Acid Detection Module (ThermoScientific), and the signal recorded on a ChemiDoc XR+ (Bio-Rad).

## Slot blot

Five female flies were mashed and the total nucleic acid phenol/Sevag extracted as described above for qPCR. Approximately 200 ng of the nucleic acid was denatured (final concentration 0.25 M NaOH, 0.5 M NaCl) for 10 min at room temperature, the sample transferred to a tube with an equal volume of ice-cold loading buffer (0.1× SSC, 0.125 M NaOH) and left on ice. The slot blotter was then prepared and samples loaded as recommended for the 48-well BioDot SF microfiltration apparatus (Bio-Rad). After loading, the wells were washed with 200 μl of loading buffer. The nylon membrane (GeneScreen Plus) was then rinsed for 2 min with 2× SSC before being UV crosslinked (Stratalinker). The membrane was first hybridized with a biotinylated rp49 RNA probe in North2-South hybridization solution (ThermoScientific) at 65°C overnight. The membrane was processed as recommended for the Chemiluminescent Nucleic Acid Detection Module (ThermoScientific), and the signal recorded on a ChemiDoc XR+ (Bio-Rad). The membrane was then stripped with a 100°C solution of 0.1× SSC/0.5% SDS (three times for ~20 min each) and re-hybridized with a *Rsp* probe (60°C overnight) and processed as above. Signals were quantitated using the ImageLab software (Bio-Rad). We determined the relative signal compared to Iso-1 for each line (5–7 replicates), and then estimate the *Rsp* copy number by scaling the relative slot blot signal to our estimate of *Rsp* copy number in Iso-1 (1100 repeats). Our Iso-1 estimate is based on *Rsp* count in a long-read assembly, which is supported by empirical slot blots (*Khost et al., 2017*).

To make the biotinylated RNA probes, gel extracted PCR amplicons (primers: *Rsp* 5′-TAATAC-GACTCACTATAGGGGAAAATCACCCATTTTGATCGC-3′ and 5′-CCGAATTCAAGTACCAGAC-3′; rp49 5′- TAATACGACTCACTATAGGGCAGTAAACGCGGTTCTGCATG-3′ and 5′-CAGCA

TACAGGCCCAAGATC-3′) were transcribed using the Biotin RNA Labeling Mix (Roche) and T7 polymerase (Promega).

## Data availability

Sequencing data generated in this paper are available in the NCBI Sequence Read Archive under project accession PRJNA647441. All data files and code to recreate analyses and figures are deposited in GitHub (https://github.com/LarracuenteLab/Dmelanogaster_satDNA_regulation*Wei, 2020*) and at the Dryad Digital Repository (https://doi.org/10.5061/dryad.hdr7sqvj3; *Wei et al., 2021b*).

## Acknowledgements

This work was supported by the National Institutes of Health General Medical Sciences (R35 GM119515 to AML), a Stephen Biggar and Elisabeth Asaro fellowship in Data Science to AML, and a University of Rochester Agnes M. and George Messersmith Dissertation Fellowship to XW. We thank Drs. Ching-Ho Chang, John Sproul, Cécile Courret, and Lucas Hemmer for providing feedback on the manuscript. We also thank the University of Rochester Center for Integrated Research Computing for access to computing facilities and the University of Rochester Genomics Research Center for sequencing services.

## Additional information

### Funding

| Funder | Grant reference number | Author |
|---|---|---|
| National Institutes of Health | R35 GM119515 | Amanda M Larracuente |
| University of Rochester | Stephen Biggar and Elisabeth Asaro fellowship | Amanda M Larracuente |
| University of Rochester | Agnes M. and George Messersmith Dissertation Fellowship | Xiaolu Wei |

The funders had no role in study design, data collection and interpretation, or the decision to submit the work for publication.

### Author contributions

Xiaolu Wei, Conceptualization, Resources, Software, Formal analysis, Validation, Visualization, Methodology, Writing - original draft, Writing - review and editing; Danna G Eickbush, Formal analysis, Validation, Visualization, Methodology, Writing - review and editing; Iain Speece, Formal analysis, Writing - review and editing; Amanda M Larracuente, Conceptualization, Supervision, Funding acquisition, Investigation, Project administration, Writing - review and editing

### Author ORCIDs

Xiaolu Wei (iD) https://orcid.org/0000-0001-9952-3757
Amanda M Larracuente (iD) https://orcid.org/0000-0001-5944-5686

### Decision letter and Author response

Decision letter https://doi.org/10.7554/eLife.62375.sa1
Author response https://doi.org/10.7554/eLife.62375.sa2

## Additional files

### Supplementary files

• Supplementary file 1. List of datasets used in this paper.

• Supplementary file 2. Count in RPM values of *Rsp/1.688* transcripts in total and poly-A RNA-seq datasets from various tissues.

- Supplementary file 3. Slot blot estimate of *Rsp* copy number.

- Supplementary file 4. *Rsp* expression level correlates with its copy number in the genome. Estimates from slot blot, northern blot, qPCR, and qRT-PCR indicate correlation between *Rsp* genomic copy number and expression level.

- Supplementary file 5. RIP-seq results for Piwi, Aub, and Ago3 from ovaries.

- Supplementary file 6. Log2 fold changes of small RNA abundance for satDNAs and piRNA clusters normalized to miRNA abundance in Rhino, Deadlock, and Cutoff (RDC) and Moon mutants.

- Supplementary file 7. Log2 fold changes of small RNA abundance for satDNAs and piRNA clusters normalized to the *flamenco* piRNA cluster in Rhino, Deadlock, and Cutoff (RDC) and Moon mutants.

- Supplementary file 8. Log2 fold change of total RNA abundance for satDNAs and piRNA clusters in Rhino, Deadlock, and Cutoff (RDC) and Moon mutants.

- Supplementary file 9. Log2 fold change of total and polyA selected RNA abundance in *rhi* mutant for satellite DNAs (satDNAs) and piRNA clusters.

- Supplementary file 10. Log2 fold change of H3K9me3 ChIP/input enrichment levels in *piwi* germline knockdown/mutant for satellite DNAs (satDNAs) and piRNA clusters.

- Supplementary file 11. *Rsp* probe sequences for RNA fluorescence in situ hybridization (FISH).

- Transparent reporting form

## Data availability

Sequencing data generated in this study have been deposited in NCBI Sequence Read Archive (SRA) under project accession PRJNA647441. Published sequencing data used in this study are from NCBI SRA database, and the full list of accession numbers can be found in Supplementary File 1.

The following dataset was generated:

| Author(s) | Year | Dataset title | Dataset URL | Database and Identifier |
|---|---|---|---|---|
| Wei X, Eickbush DG, Speece I, Larracuente AM | 2020 | Heterochromatin-dependent transcription of satellite DNAs in the *Drosophila melanogaster* female germline | https://www.ncbi.nlm.nih.gov/bioproject/PRJNA647441/ | NCBI BioProject, PRJNA647441 |

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
