## [Decision Letter]

Thank you for submitting your article "Heterochromatin-dependent transcription of satellite DNAs in the *Drosophila melanogaster* female germline" for consideration by *eLife*. Your article has been reviewed by 2 peer reviewers, and the evaluation has been overseen by Michael Eisen as the Senior and Reviewing Editor. The reviewers have opted to remain anonymous.

The reviewers have discussed the reviews with one another and the Reviewing Editor has drafted this decision to help you prepare a revised submission.

Summary:

Wei and colleagues present evidence that *Drosophila* DNA satellite repeats are transcribed via the heterochromatin-dependent piRNA source locus transcription machinery. This finding is surprising and raises the interesting question of which (if any) function such satellite expression may serve. While the paper does not provide much new mechanistic insight, it uncovers a new angle of piRNA biology. The main conclusions are well supported by the presented analyses.

Many of the analyses involve re-analysis of various RNA datasets; as always it is not possible to fully test the authors' conclusions without reanalyzing the data oneself. However, the methods appear to be appropriate for the challenges of analyzing repetitive sequences, and the 'controls' that recapitulate previous findings concerning piRNA clusters provide confidence.

The evidence that the piRNA pathway is 'important for establishing heterochromatin' is less clear. The major evidence is that H3K9me2 levels are reduced in Piwi mutants, which surprisingly is left for a supplemental figure (F2, S5). The key question is whether satellite piRNA production and the observed effects on H3K9 matter biologically. DNA FISH visualization of the satellites in piRNA regulatory mutants would help address this. It's fair to point out that many of the initial high profile papers about piRNAs and TEs only showed effects on transcript levels of TEs, but for TEs there is a plausible consequence of transcriptional derepression, namely transposition. For satellites, it's still not clear. We recognize, however, that experiments that address this question directly are challenging and are likely beyond the scope of this work. Such additional experiments done in the future can, as described above, be considered for a "Research Advance" publication.

Importantly, the reviewers and editor found the presentation in the manuscript to be disjointed and confusing, and the major focus of the revision should be to better convey the' central message of the paper by being more focused. This included the setup of the question, the organization of the figures, which often were very hard to follow, and the discussion. Some specific points to address are given below, but the authors should not limit their revision of the manuscript to these:

1. The Discussion section lacks focus. The main topic is satDNA transcription, but the discussion quickly drifts into a section on potential drivers of rapid piRNA pathway evolution, which is interesting, but disconnected to the presented data and conclusions. I would recommend to let the Discussion section have its main focus on satDNA transcription and the connected open question of the potential function thereof. A (few) hypothesis could be presented, which would also help highlight why the presented findings are important.

2. More attention could be paid to Usakin et al. They didn't analyze piRNAs but did show that a piRNA regulatory mutation (spnE) affects transcript levels from both strands and chromatin state of 1.688 satellite. They also noted differences among subfamilies.

3. It was difficult to follow what subfamily (or families) of the 1.688 satellite are being detected in various assays. Most mapping appears to have been done to the Chang/Larracuente 2019 genome. On p. 7 it is implied that this mapping only detects the 260 bp satellite, but then Figure 1, S2B is described as detecting 'all' 1.688 satellites – does this include the 359 bp and other subfamilies? What subfamily(s) does the RNA FISH detect? Some of the piRNA analyses were done using mapping to a 359 monomer, so there is evidence that this subfamily produces RNA.

---

## [Author Response]

Summary:Wei and colleagues present evidence that *Drosophila* DNA satellite repeats are transcribed via the heterochromatin-dependent piRNA source locus transcription machinery. This finding is surprising and raises the interesting question of which (if any) function such satellite expression may serve. While the paper does not provide much new mechanistic insight, it uncovers a new angle of piRNA biology. The main conclusions are well supported by the presented analyses.Many of the analyses involve re-analysis of various RNA datasets; as always it is not possible to fully test the authors' conclusions without reanalyzing the data oneself. However, the methods appear to be appropriate for the challenges of analyzing repetitive sequences, and the 'controls' that recapitulate previous findings concerning piRNA clusters provide confidence.The evidence that the piRNA pathway is 'important for establishing heterochromatin' is less clear. The major evidence is that H3K9me2 levels are reduced in Piwi mutants, which surprisingly is left for a supplemental figure (F2, S5). The key question is whether satellite piRNA production and the observed effects on H3K9 matter biologically. DNA FISH visualization of the satellites in piRNA regulatory mutants would help address this. It's fair to point out that many of the initial high profile papers about piRNAs and TEs only showed effects on transcript levels of TEs, but for TEs there is a plausible consequence of transcriptional derepression, namely transposition. For satellites, it's still not clear. We recognize, however, that experiments that address this question directly are challenging and are likely beyond the scope of this work. Such additional experiments done in the future can, as described above, be considered for a "Research Advance" publication.

We thank the reviewers and editors for their constructive feedback. We do think that our work adds valuable and novel perspective to the piRNA field. We agree that the precise ‘functions’ of these transcripts are less clear for satellites, and that we’d need to do more experiments to understand the consequences of disrupting piRNA/Piwi-mediated satDNA heterochromatin in early embryos. We have taken steps towards explicitly testing these hypotheses by manipulating RNAs from a representative complex satellite, Rsp. We have begun experiments to express Rsp short hairpin RNAs with the goal of disrupting the long non-coding RNAs and potentially the piRNA precursors, as well as overexpressing Rsp piRNAs by inserting Rsp transgene into piRNA clusters. We also have plans to precisely measure the effects on chromatin with ChIP-seq and/or CUT& TAG. We agree that these experiments are beyond the scope of the current paper and would be delighted to send this work to be considered for a “Research Advance” when it is completed.

Importantly, the reviewers and editor found the presentation in the manuscript to be disjointed and confusing, and the major focus of the revision should be to better convey the' central message of the paper by being more focused. This included the setup of the question, the organization of the figures, which often were very hard to follow, and the discussion. Some specific points to address are given below, but the authors should not limit their revision of the manuscript to these:

Again, we thank you all for your constructive feedback. This revision includes a major re-write of the discussion, reorganization of the results, and updates to the introduction and figures.

Briefly, we performed new analyses based on reviewers’ comments, including adding controls and analyzing new datasets. We also clarified our analysis of the 1.688 repeats and made sure that analyzing these repeats in different ways did not affect our conclusions.

To address issues with the presentation, we re-organized the results and figures. For figure 1, we moved the strandedness analysis for satDNA transcription from Figure 1-supplemental 2 to Figure 2-supplemental 4 in the section named “SatDNA transcription resembles dual-strand piRNA clusters” on page 11. As suggested, we added a section focusing on H3k9me3 changes in piwi embryonic knockdown ovaries on page 16, and moved the related figure out of the supplement to the main manuscript Figure 4.

We rewrote the conclusion section to focus on the central message surrounding satDNA transcription, regulation by piRNA pathways, and its involvement in heterochromatin establishment in the embryo. We hope that you agree that these revisions greatly improved the paper.

We address each point below.

1. The Discussion section lacks focus. The main topic is satDNA transcription, but the discussion quickly drifts into a section on potential drivers of rapid piRNA pathway evolution, which is interesting, but disconnected to the presented data and conclusions. I would recommend to let the Discussion section have its main focus on satDNA transcription and the connected open question of the potential function thereof. A (few) hypothesis could be presented, which would also help highlight why the presented findings are important.

This is very helpful feedback. We agree that the discussion drifted into topics that are not directly related to the results that we presented in the paper. We re-wrote the discussion to focus on satDNA transcription and regulation, rather than evolution of the piRNA pathway. We do still briefly mention implications for genome evolution as it pertains to satDNAs specifically, because we think that this is important.

2. More attention could be paid to Usakin et al. They didn't analyze piRNAs but did show that a piRNA regulatory mutation (spnE) affects transcript levels from both strands and chromatin state of 1.688 satellite. They also noted differences among subfamilies.

Yes, this is a very good point and we are grateful for the suggestion. We now include more discussion of Usakin et al. The comparisons between subfamilies in the Usakin paper were based on primer sequences for each subfamily and the sizes of the amplicons. As noted by the authors, the primers are not strictly specific to the subfamilies. One thing that we have learned from assembling these complex satellites is that what we call subfamilies at the monomer level, don’t always correspond to discrete loci in the genome (e.g. the different monomer types are often interleaved within a genomic region – e.g. 353, 353, 353, 356, 356, 353, 353 etc….). We think that analyzing based on genomic loci is more relevant to our study because these satDNA loci function like discrete piRNA clusters. When defining satDNA subfamilies based on discrete loci, we didn’t notice any major differences among the 1.688 subfamilies and our conclusions are the same as when we lump all subtypes from across the genome into one category.

We expand on this point below.

3. It was difficult to follow what subfamily (or families) of the 1.688 satellite are being detected in various assays. Most mapping appears to have been done to the Chang/Larracuente 2019 genome. On p. 7 it is implied that this mapping only detects the 260 bp satellite, but then Figure 1, S2B is described as detecting 'all' 1.688 satellites – does this include the 359 bp and other subfamilies? What subfamily(s) does the RNA FISH detect? Some of the piRNA analyses were done using mapping to a 359 monomer, so there is evidence that this subfamily produces RNA.

We regret that we were unclear about 1.688 subtypes, thank you for pointing this out. We now clarify exactly what we are doing for each analysis in the manuscript (Materials and methods, legends, and main text where it is appropriate). In general, because of the difficulty in confidently assigning reads to particular subfamily (e.g. 260-bp, 359-bp, 353-bp, 356-bp), we report the combination of 1.688 subfamilies.

For analyses where the context of the genome assembly is important (one of the strandedness analyses in Figure2—figure supplement4B), we only use satellites whose loci we had previously assembled and validated for structural features. For 1.688, that is the 260-bp satellite. This repeat unit has a large internal deletion compared to other 1.688, which made assembling and distinguishing this repeat more tractable. For the analysis of piRNA read distribution along satellite consensus (Figure2—figure supplement2), we used reads that mapped to all 1.688 subfamilies and then differentiated these reads by blasting to the different subfamilies to get the distribution along each consensus (e.g. 359-bp and 260-bp). We now explain these details on page 11-12.

For the ChIP-seq analyses, we think it is important to consider satDNA as discrete loci, rather than repeat unit types. For these analyses, we excluded the euchromatic 1.688 loci as stated in the figure legend and MATERIAL AND METHODS (page 30) and only consider reads that map uniquely to heterochromatic loci (note that our conclusions do not change if we consider all mapped reads instead of just uniquely mapping for this analysis; see “ different.1pt688.anaylsis” in https://github.com/LarracuenteLab/Dmelanogaster_satDNA_regulation; Wei 2020). The heterochromatic loci we analyzed are: the 2L 260-bp locus (2L_2: 402701-460225), the 3L 353/356-bp locus (3L_3: 46695-106272), and unmapped loci on Contig101 and Contig9, which are a mix of 353/356-bp and 359-bp. Our conclusions do not change if we are agnostic about where these repeats come from in the genome and combine all 1.688 subfamilies (see “ different.1pt688.anaylsis” in https://github.com/LarracuenteLab/Dmelanogaster_satDNA_regulation; Wei 2020), or if we categorize each 1.688 subfamily based on similarity to a consensus repeat unit (see “ different.1pt688.anaylsis” in github https://github.com/LarracuenteLab/Dmelanogaster_satDNA_regulation; Wei 2020). We present the results for the locus analysis for the ChIP-seq analysis because we think that this makes the most sense. The full analyses are in Github and deposited in the Dryad (doi forthcoming).

For all other analyses (total and small RNA-seq, RIP-seq), we originally reported result for each subfamily of 1.688, and we now report the combination of the subfamilies. The conclusions remain the same (see “ different.1pt688.anaylsis” in github https://github.com/LarracuenteLab/Dmelanogaster_satDNA_regulation; Wei 2020).

For the RNA FISH in Figure1B and Figure 1—figure supplement 2, the 1.688 probe that we use recognizes all of the 1.688 subfamilies except for the 260-bp on 2^nd^ chromosome i.e. 359-bp on the X chromosome, 356-bp and 353-bp on the 3^rd^ chromosome; also see. We now state this clearly in the figure legends and Materials and methods (page 31).

References:

Akkouche A, Mugat B, Barckmann B, Varela-Chavez C, Li B, Raffel R, Pelisson A, Chambeyron S. 2017. Piwi Is Required during *Drosophila* Embryogenesis to License Dual-Strand piRNA Clusters for Transposon Repression in Adult Ovaries. Mol Cell 66:411-419 e414.

Ferree PM, Barbash DA. 2009. Species-specific heterochromatin prevents mitotic chromosome segregation to cause hybrid lethality in *Drosophila.* PLoS biology 7:e1000234.

Handler D, Olivieri D, Novatchkova M, Gruber FS, Meixner K, Mechtler K, Stark A, Sachidanandam R, Brennecke J. 2011. A systematic analysis of *Drosophila* TUDOR domain-containing proteins identifies Vreteno and the Tdrd12 family as essential primary piRNA pathway factors. EMBO J 30:3977-3993.

Khost DE, Eickbush DG, Larracuente AM. 2017. Single-molecule sequencing resolves the detailed structure of complex satellite DNA loci in *Drosophila melanogaster*. Genome Res 27:709-721.

Malone CD, Brennecke J, Dus M, Stark A, McCombie WR, Sachidanandam R, Hannon GJ. 2009. Specialized piRNA pathways act in germline and somatic tissues of the *Drosophila* ovary. Cell 137:522-535.

Olivieri D, Senti KA, Subramanian S, Sachidanandam R, Brennecke J. 2012. The cochaperone shutdown defines a group of biogenesis factors essential for all piRNA populations in *Drosophila.* Mol Cell 47:954-969.

Riddle NC, Minoda A, Kharchenko PV, Alekseyenko AA, Schwartz YB, Tolstorukov MY, Gorchakov AA, Jaffe JD, Kennedy C, Linder-Basso D, et al. 2011. Plasticity in patterns of histone modifications and chromosomal proteins in *Drosophila* heterochromatin. Genome Res 21:147-163.